# Segment Anything in Optical Coherence Tomography: SAM 2 for Volumetric Segmentation of Retinal Biomarkers

**DOI:** 10.3390/bioengineering11090940

**Published:** 2024-09-19

**Authors:** Mikhail Kulyabin, Aleksei Zhdanov, Andrey Pershin, Gleb Sokolov, Anastasia Nikiforova, Mikhail Ronkin, Vasilii Borisov, Andreas Maier

**Affiliations:** 1Pattern Recognition Lab, Department of Computer Science, Friedrich-Alexander-Universität Erlangen-Nürnberg, 91058 Erlangen, Germany; andreas.maier@fau.de; 2“VisioMed.AI”, Golovinskoe Highway, 8/2A, 125212 Moscow, Russia; zhdanov@visiomed.ai (A.Z.); gleb.m.sokolov@gmail.com (G.S.); 3Engineering School of Information Technologies, Telecommunications and Control Systems, Ural Federal University Named after the First President of Russia B. N. Yeltsin, 620002 Yekaterinburg, Russia; a.d.pershin@urfu.me (A.P.); m.v.ronkin@urfu.ru (M.R.); v.i.borisov@urfu.ru (V.B.); 4Ophthalmosurgery Clinic “Professorskaya Plus”, Vostochnaya, 30, 620075 Yekaterinburg, Russia; dr.nikiforova@inbox.ru; 5Preventive and Family Medicine, Ural State Medical University, Repina, 3, 620028 Yekaterinburg, Russia

**Keywords:** OCT, segmentation, SAM, MedSAM, AMD, DME, retina

## Abstract

Optical coherence tomography (OCT) is a non-invasive imaging technique widely used in ophthalmology for visualizing retinal layers, aiding in the early detection and monitoring of retinal diseases. OCT is useful for detecting diseases such as age-related macular degeneration (AMD) and diabetic macular edema (DME), which affect millions of people globally. Over the past decade, the area of application of artificial intelligence (AI), particularly deep learning (DL), has significantly increased. The number of medical applications is also rising, with solutions from other domains being increasingly applied to OCT. The segmentation of biomarkers is an essential problem that can enhance the quality of retinal disease diagnostics. For 3D OCT scans, AI is beneficial since manual segmentation is very labor-intensive. In this paper, we employ the new SAM 2 and MedSAM 2 for the segmentation of OCT volumes for two open-source datasets, comparing their performance with the traditional U-Net. The model achieved an overall Dice score of 0.913 and 0.902 for macular holes (MH) and intraretinal cysts (IRC) on OIMHS and 0.888 and 0.909 for intraretinal fluid (IRF) and pigment epithelial detachment (PED) on the AROI dataset, respectively.

## 1. Introduction

Optical coherence tomography (OCT) is a non-invasive imaging technique that uses low-coherence interferometry to produce high-resolution, cross-sectional images [1]. In ophthalmology, OCT allows for detailed visualization of the retina, enabling early detection and continuous monitoring of various retinal diseases such as age-related macular degeneration (AMD) [2] and diabetic macular edema (DME) [3]. Deep learning (DL) has made significant breakthroughs in medical imaging, particularly for image classification and segmentation. Studies have demonstrated that using DL for interpreting OCT is efficient and accurate and performs well for discriminating eyes affected by a disease from normal [4]. This suggests that incorporating DL technology in OCT addresses the gaps in the current practice and clinical workflow [4]. Recent advances in DL algorithms offer accurate detection of complex image patterns, achieving accuracy levels in image classification and segmentation tasks. Segmentation analysis of the SD-OCT image may also eliminate the need for manual refinement of conventionally segmented retinal layers and biomarkers [5].

Modern artificial intelligence (AI) methods have recently increased exponentially due to the advent of affordable computational capabilities. Many innovative segmentation methods are generalized on widespread datasets. For example, a novel Segment Anything Model (SAM) was mainly trained on datasets of ordinary objects [6]. Although it contains microscopy [7] and X-ray [8] images, according to the original paper, it does not contain OCT or fundus images. Therefore, using such models in ophthalmology could be enhanced through fine-tuning the foundational models. Applying such innovative models in OCT can improve the quality of ophthalmologic service and assist physicians in decision-making [9].

### 1.1. Deep Learning for Segmentation of OCT Biomarkers

In recent years, the application of DL in OCT has developed enormously, and many methods for biomarkers segmentation on OCT B-scans have been developed, addressing specific challenges like the segmentation of macular holes (MH), intraretinal cysts (IRC), intraretinal fluid (IRF), and pigment epithelial detachment (PED). Table 1 summarizes novel studies in the domain.

Ganjee et al. [10] proposed an unsupervised segmentation approach in a three-level hierarchical framework for segmenting IRC regions in the SD-OCT images. In the first level, the ROI mask is built, keeping the exact retina area. The prune mask is built in the second level, reducing the search area. In the third level, the cyst mask is extracted by applying the Markov Random Field (MRF) model and employing intensity and contextual information. The authors tested their approach on the public datasets where B-scans with IRC are available: OPTIMA [19], UMN [20], and Kermany [21].

Wang et al. [14] proposed a new CNN-based architecture D3T-FCN by integrating the principle of Auto-Encoder and the U-Net model for the joint segmentation of MH and cystoid macular edema (CME) in retinal OCT images. The authors evaluated the method with the AROI dataset.

U-Net-based models are still widespread. In contemporary works, authors still use U-Net and its modifications. Rahil et al. [11] proposed an ensemble approach utilizing U-Net models with a combination of the relative layer distance, independently trained for IRF, SFR, and PED classes, available in the RETOUCH dataset. Ganjee et al. [12] proposed a two-staged framework for IRC segmentation. In the first step, prior information is embedded, and the input data are adjusted. In the second step, an extended structure of the U-Net with an implemented connection module between the encoder and decoder parts was applied to the scans to predict the masks. Melinščak et al. [13] used Attention U-Net and applied it to the AROI dataset, focusing on AMD biomarkers, particularly subretinal fluid (SRF), IRF, and PED. Attention added to U-Net architecture is a way to highlight relevant activations during the training, which potentially leads to better generalization. Daanouni et al. [15] proposed a new architecture inspired by U-Net++ and a spatially adaptive denormalization unit with a class-guided module. The segmentation was performed in two stages to segment the layers and fluids in the AROI dataset. George et al. [16] used the U-Net model trained on DME scans from Kermany dataset and evaluated the model on the Lu et al. [22] dataset.

New works pioneered using SAM-based models for ophthalmology have recently begun to appear. Qiu et al. [17] explored the Segment Anything Model (SAM) application in ophthalmology, including in their training process fundus, OCT, and OCT Angiography (OCTA) datasets. The authors proposed a fine-tuning method, training only the prompt layer and task head based on a one-shot mechanism. The model’s performance raised about 25% of the Dice score compared with the pure SAM. Fazekas et al. [18] adapted SAM specifically for the OCT domain using a fine-tuning strategy on the RETOUCH dataset and compared IRF, SRF, and PED segmentation results for pure SAM, SAM with a fine-tuned decoder, and SAMed. Fine-tuning the decoder led to a significant performance improvement, with an increase in Dice score of up to 50% compared to zero-short segmentation using SAM.

These studies collectively showcase the ongoing advancements in OCT image segmentation. Models like U-Net and its variants continue to dominate, while newer approaches like SAM are beginning to emerge, promising enhanced segmentation performance across various retinal biomarkers.

### 1.2. OCT Biomarkers Segmentation

Identifying specific biomarkers in OCT images can help classify retinal diseases such as AMD, diabetic retinopathy (DR), DME, central serous chorioretinopathy (CSH), retinal vein occlusion (RVO), retinal artery occlusion (RAO), and vitreoretinal interface disease (VID).

The modern literature uses various biomarker names to describe typical changes in the macula seen in OCT: disorganization of retinal inner layers (DRIL) [23,24], different stages of outer retina atrophy (ORA) [25], soft drusen [26], double-layer sign [27], IRF, SRF, PED [28], MH, IRC [29], etc.

With the rapid development of OCT technology, new biomarkers regularly appear, and older classifications are revised. For example, IRC and IRF are very similar. Morphologically, they both refer to cavities in the retina’s neuroepithelium of different origins. Cavities found around a MH in VID are often called IRC, while cysts formed by fluid leaking from retinal vessels in DR, DME, or AMD are more often called IRF. The names of biomarkers often vary depending on the size and contents of the cavities caused by the disease’s pathogenesis.

In public datasets, we found four biomarkers that can describe volumetric changes in the macula. IRF and PED often appeared together when describing vascular-related pathologies (AMD, DME, RVO), while MH and IRC were paired to describe VID and DR, Table 1.

Figure 1a,b show two linear images taken from the same patient in different areas of the macula. Figure 1a passes through the center of the macula, while Figure 1b passes through an area next to the main defect. Figure 1b shows multiple IRCs that form in the areas surrounding the macular hole in response to damage. Figure 1a shows the MH biomarker, which provides important diagnostic information about a full-thickness defect in the macula, including the photoreceptor layer, which can severely affect vision. In modern vitreous surgery, the maximum width of this biomarker influences the choice of surgical treatment and vision prognosis. The automated analysis of the volumetric characteristics of this defect could improve this aspect of ophthalmology.

If it were not known that the images show adjacent areas of the same patient, the IRC biomarker in Figure 1b could suggest vascular-related pathologies (AMD, DME, RVO), where fluid accumulation in the neuroepithelium is more commonly called IRF. Therefore, Figure 1a,b highlight the importance of evaluating macular tissue conditions using multiple images. In similar cases, especially with smaller macular holes, there could be a diagnostic error and incorrect treatment.

Figure 1c illustrates IRF, or fluid accumulation within the retina, which may signal inflammatory changes or the result of neovascularization [30]. Figure 1d demonstrates PED, which involves the detachment of the retinal pigment epithelium from the underlying choroid, commonly associated with AMD, DR, or CSH [31].

It is important to accurately understand the volumetric characteristics of IRF and PED to assess disease stages, monitoring frequency, when to start invasive procedures, and choosing the right medications and their dosing schedule. Their detection and accurate segmentation in OCT images are essential for effective disease management and reducing the risk of blindness.

In this paper, we adapt the SAM 2 using MedSAM 2 for the OCT domain and examine the performance measuring IRC, IRF, PED, and MH biomarkers–the most important biomarkers for managing patients with macular, diabetic, and vascular retinal diseases. We evaluate different SAM 2 modalities on public datasets [32,33]. We apply modalities like point and box selections with fine-tuned MedSAM 2 for the volumetric segmentation of 3D OCT and compare the metrics with pure SAM 2 and the classical U-Net model.

## 2. Methods

### 2.1. Segment Anything Models

SAM is a cutting-edge segmentation model that handles various segmentation tasks [6]. SAM can perform zero-shot segmentation, which means it can segment objects in images without requiring task-specific training or fine-tuning. SAM architecture combines a robust image encoder, prompt encoder, and mask decoder. In the original work, the image encoder was an MAE [34] pre-trained Vision Transformer (ViT) [35]. A prompt encoder has two sets of prompts: sparse (points, boxes, text) and dense (masks). Points and boxes are represented by positional encodings summed with learned embeddings for each prompt type and free-form text with an off-the-shelf text encoder from CLIP [36]. Dense prompts are embedded using convolutions and summed element-wise with the image embedding. The image encoder outputs an image embedding that can be efficiently queried by various input prompts to produce the object masks. SAM can output multiple valid masks with corresponding confidence scores. Figure 2 shows the model architecture adapted for the OCT domain.

SAM achieved sufficient segmentation results primarily on the objects characterized by distinct boundaries. However, SAM has significant limitations in segmenting typical medical images with weak boundaries or low contrast.

MedSAM [37] was proposed to overcome these limitations in the medical domain: this is a refined foundation model that improves the segmentation performance of SAM on medical images. MedSAM accomplishes this by fine-tuning SAM on a specific medical dataset with more than one million medical image-mask pairs. The MedSAM was initialized with the pre-trained SAM ViT-Base. The prompt encoder was frozen during the training since it could already encode the bounding box prompt. All the trainable parameters in the image encoder and mask decoder were updated. MedSAM was trained with a large-scale medical image dataset containing over 1.5 million image-mask pairs covering ten imaging modalities and over 30 cancer types.

The first generation of SAM was designed to work with 2D images. Of course, using 3D images is possible with frame-by-frame analysis, but there is no connection between the images. Updated SAM 2 [38] can produce segmentation masks of the object of interest across video frames. The model has a memory that stores information about the object from previous frames, which allows it to generate masks throughout the video and correct them based on the object’s stored memory context. This architecture generalizes SAM to the video domain and could also potentially be applied to 3D data. Figure 3 shows SAM 2 architecture adapted for OCT volumes.

MedSAM 2 [39] adopted the philosophy of taking medical images as videos, adapting SAM 2 for the 3D medical data. Based on SAM 2, MedSAM 2 introduced an additional confidence memory bank and weighted pick-up strategy. In SAM 2, we merge all information equally when combining a new input image embedding with data from the memory bank. Alternatively, MedSAM 2 uses a weighted pick-up strategy to assign higher weights to images that are more similar to the input image. During training, MedSAM 2 employs a calibration head to ensure that the model associates higher confidence with more accurate segmentations and lower confidence with less accurate ones. This calibration aligns the model’s confidence with the accuracy of its predictions, improving the effectiveness of the confidence memory bank.

Thus, in this work, we applied MedSAM 2 to adapt SAM 2 for the segmentation of biomarkers in OCT volumes.

### 2.2. U-Net Model

The U-Net architecture [40], initially proposed for biomedical segmentation problems, is a symmetric network with an encoder and a decoder connected by skip connections that efficiently combine low-level and high-level features to reconstruct spatial information accurately. In the original version of U-Net, the encoder successively reduces the image size with increasing feature depth, and the decoder restores image resolution using transposed convolutions. In this work, a modified version doubled the number of channels in the bottleneck layer, increasing the feature depth and helping to represent complex objects better. Here, we used the classical U-Net model as a baseline because, according to the number of publications, it is one of the most popular models for OCT biomarker segmentation.

### 2.3. Datasets

To evaluate the quality of model segmentation, we used datasets including labeled OCT volumes, in which there is a series of frame-by-frame OCT images per patient’s eye. The AROI dataset [32] consists of macular SD-OCT volumes recorded with the Zeiss Cirrus HD-OCT 4000 device, Carl Zeiss AG, Germany. Each OCT volume consists of 128 B-scans with a 1024×512 pixels resolution and a pixel size of 1.96×11.74 μm. The dataset includes OCT volumes of 24 patients diagnosed with AMD undergoing active anti-VEGF therapy. The dataset has masks with eight classes: above internal limiting membrane (ILM); ILM—inner plexiform layer (IPL)/inner nuclear layer (INL); IPL/INL—retinal pigment epithelium (RPE); RPE—Bruch membrane (BM); under BM; PED; SRF; IRF. In total, the authors annotated 1136 B-scans of 24 patients, providing an average of 47 annotated B-scans per one 3D volume.

The OIMHS dataset [33] comprises 125 OCT volumes from 125 eyes, having 3859 OCT B-scans from 119 patients with MH. OCT volumes were performed in all patients using the SD-OCT system (Spectralis HRA OCT, Heidelberg Engineering, Heidelberg, Germany). All the B-scans have 512×512 pixels resolution. The authors annotated each B-scan and provided the masks with four classes: MH, IRC, retina, and choroid. On average, there are 30 B-scans per one 3D volume.

### 2.4. Training

Given an input OCT volume $X\in R^{H\times W\times N}$, where *H* and *W* are the height and width of the frame, respectively, and *N* is the number of frames in the volume, we aim to obtain a segmentation mask $Y\in R^{H\times W\times N\times C}$, where *C* is the number of classes.

We split the volumes from the AROI and OIMHS datasets into training and test subsets at the patient level with a training–test ratio of 80:20 for both datasets, such that all dataset biomarkers were performed in both subsets.

For the training, we used the pre-trained SAM 2 weights “sam2_hiera_small.pt” available at the SAM 2 official repository [38]. The input image_size was set to 1024. The training was conducted on an NVIDIA A100 (40 GB) GPU using a conda virtual environment with Python 3.12, CUDA 12.1, and PyTorch 2.4.0. Training on 100 epochs on one dataset takes up to 3 h. The training loss on AROI dataset was 0.0298, and 0.0653 on OIMHS dataset.

For the training of the U-Net model, we applied the Adam optimizer, categorical cross-entropy loss function, and ReLU activation function. The batch size was set to 8, contributing to the efficient use of computing resources and speeding up the training process. The learning rate was $10^{-4}$. To work with GPUs, the scaler GradScaler was chosen. The model training was carried out on 2× Nvidia Tesla T4 GPUs. For the AROI dataset, the average loss was 0.0219, and for the OIMHS dataset, the average loss was 0.0495.

The inference of all the models in this study was carried out on a local machine with NVIDIA GeForce RTX 3070 GPU and AMD Ryzen 9 5900HX × 16 processors.

### 2.5. Metrics

For the model evaluation, we applied two of the most popular metrics for segmentation tasks: Intersection over Union (IoU), also known as the Jaccard index, and Dice score, also known as the Dice Similarity Coefficient (DSC) or the F1 score. IoU is calculated by dividing the intersection between the predicted (A) and ground truth (B) regions by the area of their union:(1)$$IoU\left(A,B\right)=\frac{\left|A\cap B\right|}{\left|A\cup B\right|}=\frac{TP}{TP+FP+FN},$$
where TP=TruePositive, FP=FalsePositive, and FN=FalseNegative. The Dice score equals twice the size of the intersection of predicted (A) and ground truth (B) regions divided by their sum:(2)$$DSC\left(A,B\right)=\frac{2|A\cap B|}{\left|A|+|B\right|}=\frac{2\times TP}{2\times TP+FP+FN}.$$

Dice score and IoU are the most commonly used metrics for semantic segmentation as they penalize the FP, a common factor in highly class-imbalanced datasets [41].

## 3. Results

For the evaluation, we compared different SAM 2 prompt modes with the results of the segmentation using U-Net for four types of biomarkers separately. The selected biomarkers were MH, IRC, IRF, and PED. Table 2 shows the IoU and Dice scores for all of the experiments. Fine-tuned on OCT data, MedSAM 2 models show better IoU and Dice scores than U-Net for most biomarker types. Also, the results showed that using box selection is more efficient than point selection, although the opposite was true for IRF. Table 2 shows the superiority of MedSAM 2 in the box selection mode compared to U-Net by 8.2–9.2% in IoU and 9.5% in Dice score. MedSAM 2 with box selection mode is better than U-Net, with a 5.7% improvement in the IoU score and a 3.3% improvement in the Dice score. In the case of the IRF biomarker, MedSAM 2 with point selection mode achieves the best result, outperforming U-Net by 5% and 2.8% in the IoU and Dice scores, respectively.

Figure 4 shows the performance of MedSAM 2 segmentation modes on the OIMHS dataset, displaying IRC and MH segmentation examples, which are shown in blue and red, respectively. Figure 5 shows the performance of MedSAM 2 segmentation modes on the AROI dataset, displaying PED segmentation examples, shown in yellow.

Point selection modality is performed via selection of the related to the region points of interest on a single OCT image slice, as shown in Figure 4d,i and Figure 5d. Box selection modality is carried out by selecting the related square area where the biomarker is located, as shown in Figure 4e,j and Figure 5e. The segmentation of the corresponding biomarkers is then carried out on all OCT slices.

Figure 6 demonstrates the volumetric segmentation of IRC using the point selection modality on the scans from the OIMHS dataset. In the example, point selection was carried out with three positive prompts shown in green and two negative prompts shown in red. IRC mask predictions are shown in blue.

Similarly, Figure 7 shows the volumetric segmentation of PED using the point selection modality on scans from the AROI dataset. In the example, point selection was carried out with one positive prompt shown in green. PED mask predictions are shown in orange.

These examples illustrate the model’s ability to perform high-quality segmentation across various OCT image slices, ensuring the accurate identification of regions of interest, specifically the PED and IRC biomarkers.

Apart from evaluating the model on the mentioned datasets, we also tested the model trained on the OIMHS on the OCTDL dataset [9], specifically on B-scans from the VID class with MH. Figure 8 shows examples of experiments with box selection modality. Since the definition of ground truth for the upper boundary of the holes themselves is controversial, a medical expert visually assessed the quality of the predicted MH masks. For the evaluation of the IRC biomarker, we used 76 manually annotated B-scans and obtained an IoU score of 0.811 and a Dice score of 0.891. These results are consistent with the performance observed on the OIMHS dataset, suggesting that the model is capable of generalizing across different data sources.

## 4. Discussion

The metrics obtained from MedSAM 2 are higher than those from U-Net for all classes of biomarkers used in the evaluation. The inclusion of spatial information improves the quality of segmentation. The model has a wide range of applications. Aside from automated segmentation, it can also be used in medical data labeling applications. This model will significantly reduce the time required for a medical worker to label specific biomarkers, unlike U-Net, which does not offer real-time supervision. This is especially useful for 3D segmentation, such as when a medical worker needs to label the same biomarker throughout an entire 3D scan volume.

Exploring the model’s potential on diverse datasets, such as the OCTDL dataset categorized by pathological conditions, could open up new research and application routes. In this scenario, images with similar biomarkers could be used instead of the frames of the volume scan, offering a new perspective of the memory bank and the model’s capabilities.

In modern ophthalmologic practice, many classic and contemporary biomarkers are highlighted to help the practitioner identify the correct macular disease, choose tactics, and predict treatment outcomes. Often, such marking is manual and time-consuming; some information may be missed due to the abundance of scans. From the parameters automated for analysis in modern tomography software available in the broad world of ophthalmic practice, single biomarkers (central retinal thickness, avascular area) are used, obtained through summing up linear measurements on thousands of B-scan images. Volumetric measures (MH, IRF, IRC, PED, etc.) are not automatically analyzed. However, this information could make it easier and more accurate for ophthalmologists to combat macular diseases, the leading causes of blindness worldwide.

## 5. Conclusions

The rapidly advancing field of OCT imaging has seen significant progress with DL methods. U-Net and its variants have been widely used due to their reliable performance in segmenting OCT biomarkers, such as MH, IRC, and PED. However, recent developments, like SAM and its medical adaptations, particularly MedSAM 2, are beginning to outperform U-Net models.

SAM 2, specifically designed for the labeling of spatial data, introduces an advanced approach with memory-based learning to enhance segmentation accuracy that can be employed for the segmentations of medical volumetric scans. With the availability of several prompt modes, SAM-based architectures allow the control of the segmentation process and are promising to be utilized in semi-automatic labeling tools for medical data.

Evaluations across public datasets reveal that fine-tuned MedSAM 2, specifically designed for medical data, generally outperforms U-Net, especially in scenarios requiring the detailed segmentation of biomarkers. Thus, MedSAM 2 is a promising tool for segmenting OCT biomarkers, especially when segmenting 3D data is necessary. 

## Figures and Tables

**Figure 1 bioengineering-11-00940-f001:**
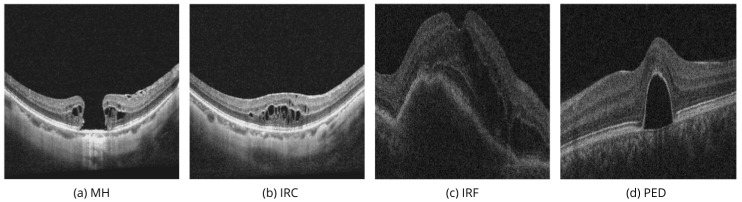
OCT image segmentation biomarkers. Examples of macular hole (MH) (**a**) and intraretinal cysts (IRC) (**b**) from OIMHS dataset. Examples of intraretinal fluid (IRF) (**c**) and pigment epithelial detachment (PED) (**d**) from the AROI dataset.

**Figure 2 bioengineering-11-00940-f002:**
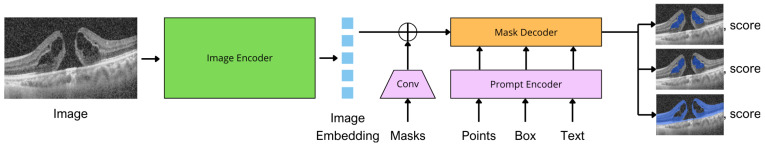
Segment Anything Model (SAM) overview. Example of OCT B-scan segmentation with different prompts. The image encoder maps the input image into a high-dimensional image embedding space. The prompt encoder transforms the user’s prompts into feature representations. The mask decoder fuses the image embedding and prompts the masks using cross-attention. Adapted from [6].

**Figure 3 bioengineering-11-00940-f003:**
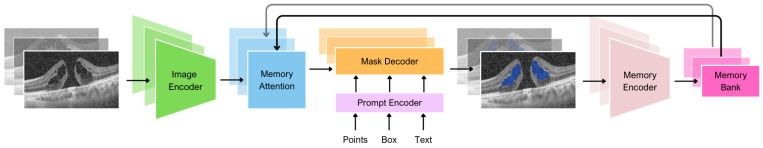
Segment Anything Model 2 (SAM 2) overview. The segmentation prediction is conditioned on the current prompt and previously observed memories for a given volume. The image encoder consumes frames one at a time and cross-attends to memories of the target object from previous frames. The mask decoder predicts the segmentation mask for the frame. The memory encoder transforms the prediction and image encoder embeddings for future usage. Adapted from [38].

**Figure 4 bioengineering-11-00940-f004:**
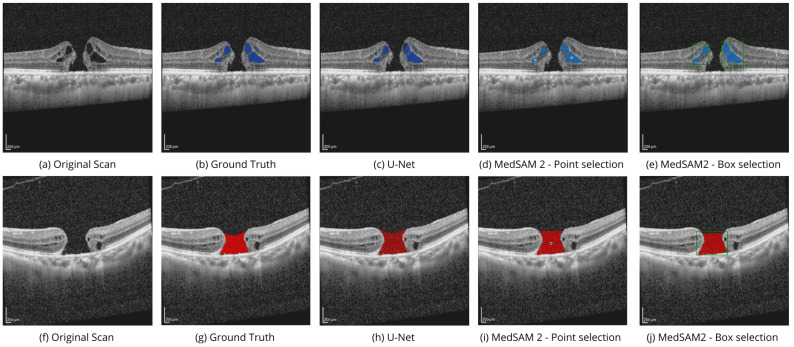
Performance of SAM 2 segmentation modes on the OIMHS dataset. Original OCT scans (**a**,**f**) with available ground truth masks (**b**,**g**) from the test subset were processed with the U-Net model (**c**,**h**) and MedSAM 2 in two modes: point selection (**d**,**i**) and box selection (**e**,**j**). The images show IRC and MH segmentation examples, which are shown in blue and red, respectively.

**Figure 5 bioengineering-11-00940-f005:**
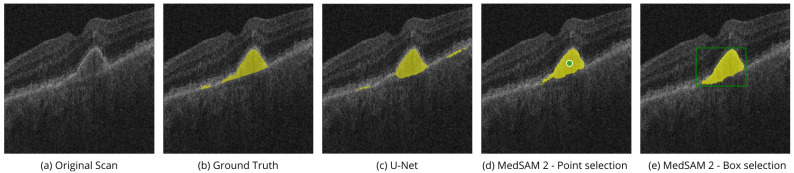
Performance of SAM 2 segmentation modes on the AROI dataset. Original OCT scans (**a**) with available ground truth masks (**b**) from the test subset were processed with the U-Net model (**c**) and MedSAM 2 in two modes: point selection (**d**) and box selection (**e**). The images show PED segmentation example, which is shown in yellow.

**Figure 6 bioengineering-11-00940-f006:**
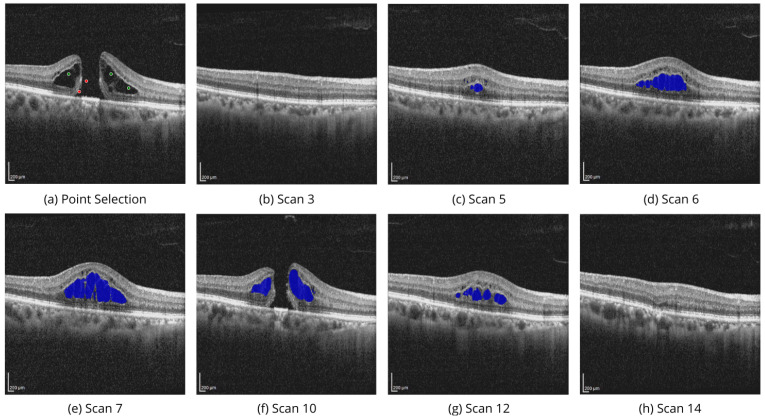
Volumetric segmentation of IRC using point selection modality on OIMHS dataset. In this example, point selection was carried out with three positive prompts shown in green and two negative shown in red color (**a**). IRC mask predictions on slices (**b**–**h**) are shown in blue.

**Figure 7 bioengineering-11-00940-f007:**
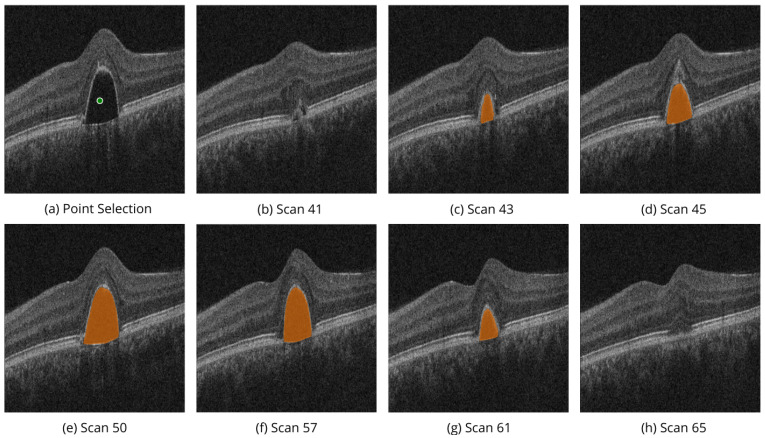
Volumetric segmentation of PED using point selection modality on AROI dataset. In this example, point selection was carried out with one positive prompt shown in green (**a**). PED mask predictions on slices (**b**–**h**) are shown in orange.

**Figure 8 bioengineering-11-00940-f008:**
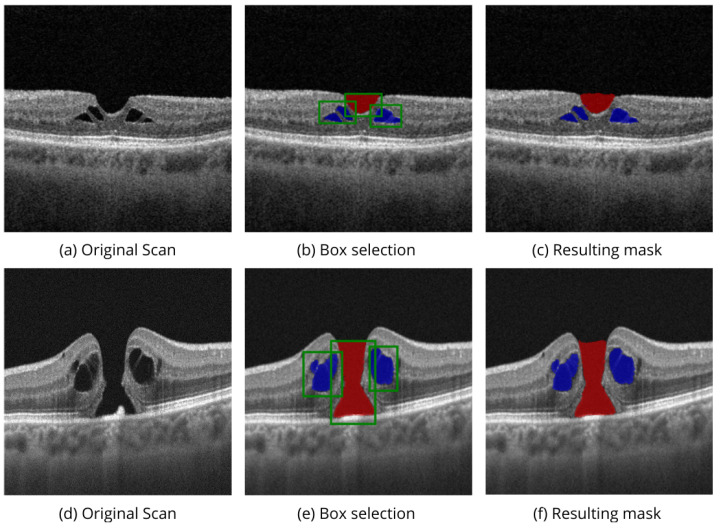
Segmentation of IRC and MH biomarkers using box selection modality on OCTDL dataset. Box selection mode was applied on original scans (**a**,**d**) with multiple boxes (**b**,**e**). Red and blue colors on predicted masks (**c**,**f**) correspond to MH and IRC accordingly.

**Table 1 bioengineering-11-00940-t001:** Overview of deep learning methods for segmentation of OCT biomarkers.

Year	Author	Dataset	MH ^1^	IRC ^2^	IRF ^3^	PED ^4^	Diseases	Model
2020	Ganjee [10]	OPTIMA, UMN, Kermany	No	Yes	No	No	AMD, DME	Markov Random Field
2023	Rahil [11]	RETOUCH	No	No	Yes	Yes	AMD, DME, RVO	U-Net ensemble
2023	Ganjee [12]	OPTIMA, UMN, Kermany	No	Yes	No	No	AMD, DME	Modified U-Net
2023	Melinščak [13]	AROI	No	No	Yes	Yes	AMD	Attention-based U-Net
2023	Wang [14]	AROI	Yes	Yes	No	No	MH, DR	D3T-FCN
2023	Daanouni [15]	AROI	No	No	Yes	Yes	AMD	U-Net++
2024	George [16]	Kermany	No	No	Yes	No	DME	U-Net
2024	Qiu [17]	AROI	No	No	Yes	No	AMD	SAM
2024	Fazekas [18]	RETOUCH	No	No	Yes	Yes	AMD, DME, RVO	SAM, SAMed

^1^ Macular hole. ^2^ Intraretinal cysts. ^3^ Intraretinal fluid. ^4^ Pigment epithelial detachment.

**Table 2 bioengineering-11-00940-t002:** Evaluation of segmentation performance on the volumetric OCT scans from OIMHS and AROI datasets. IoU and Dice score were computed for MH, IRC, IRF, and PED classes. Best results are marked in bold.

**Experiment**	**OIMHS**				**AROI**			
	**MH**		**IRC**		**IRF**		**PED**	
	**IoU ^1^**	**Dice**	**IoU**	**Dice**	**IoU**	**Dice**	**IoU**	**Dice**
SAM 2—Point Selection	0.201	0.335	0.109	0.196	0.172	0.293	0.102	0.185
SAM 2—Box Selection	0.214	0.352	0.113	0.203	0.175	0.298	0.112	0.201
U-Net	0.771	0.871	0.762	0.865	0.759	0.863	0.784	0.879
MedSAM 2—Point Selection	0.814	0.897	0.827	0.906	**0.799**	**0.888**	0.809	0.895
MedSAM 2—Box Selection	**0.840**	**0.913**	**0.821**	**0.902**	0.791	0.884	**0.832**	**0.909**

^1^ Intersection over Union.

## Data Availability

The data used in this study are available in the public domain [32,42].

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
