# Peer review of "Segment Anything in Optical Coherence Tomography: SAM 2 for Volumetric Segmentation of Retinal Biomarkers"

_bioengineering, 2024, doi:10.3390/bioengineering11090940_

Round 1
Reviewer 1 Report
Comments and Suggestions for Authors
In this study the authors used the new SAM 2 and MedSAM 2 the segmentation of OCT volumes for two open-source datasets, comparing their performance with the traditional U-Net and found that the model achieved an overall Dice score of 0.913 and 0.902 for macular holes (MH) and intraretinal cysts (IRC) on OIMHS and 0.888 and 0.909 for intraretinal fluid (IRF) and pigment epithelial detachment (PED) on the AROI dataset, respectively. Some concerns and suggestions are listed as below:
These results should be verified using another cohort (apart from open-source datasets).
How about using a combination of biomarkers in this study?
What do you mean by saying 'Fine-tuned MedSAM 2 models are better than U-Net in most cases'?
"The results show that" should be "The results showed that" in line 242.
How about the number of OCT slices in this study? A large number of OCT slices may have different effects on final results.
I wonder if OCT slices used in this study were tested in a similar machine.
Comments on the Quality of English Language
The English of this manuscript should be edited.
Author Response
We are grateful for the review of our manuscript. Responses to each comment are provided in the attached file.

Reviewer 2 Report
Comments and Suggestions for Authors
The manuscript titled "Segment Anything in OCT: SAM 2 for Volumetric Segmentation of Retinal Biomarkers" discusses the use of Optical Coherence Tomography (OCT) for imaging the retina to detect retinal diseases like age-related macular degeneration (AMD) and diabetic macular edema (DME). The paper focuses on employing the Segment Anything Model (SAM 2) and MedSAM 2 for automated volumetric segmentation of retinal biomarkers from 3D OCT scans. The authors compare the performance of these models with U-Net, showing that MedSAM 2 achieved superior segmentation results, making it a promising tool for retinal disease diagnostics by enhancing the accuracy and reducing the manual labor involved in OCT biomarker segmentation.
The manuscript is organized and well-written. I have only minor concerns:
1. The title should be modified to: "Segment Anything Model in Optical Coherence Tomography: SAM 2 for Volumetric Segmentation of Retinal Biomarkers".
2. The titles of tables should be above (not below) the tables. This applies to tables only (not figures).
3. All abbreviations that are included in the tables and figures should be defined in the footnotes of the tables or the figure legends.
Author Response

(The authors gave the same response as above.)

Round 2
Reviewer 1 Report
Comments and Suggestions for Authors
The authors have addressed my previous concerns.
Comments on the Quality of English Languagefine